# Silica and Biochar Amendments Improve Cucumber Growth under Saline Conditions

**Manar Al-Toobi** [1,2], **Rhonda R. Janke** [1], **Muhammad Mumtaz Khan** [1], **Mushtaque Ahmed** [3], **Waleed M. Al-Busaidi** [1] and **Abdul Rehman** [1,4,*]

1   Department of Plant Sciences, College of Agricultural and Marine Sciences, Sultan Qaboos University, Al-Khoud 123, Oman; manar.altobi@ea.gov.om (M.A.-T.)
2   Azzan Bin Qais International School, Muscat 102, Oman
3   Soil, Water & Agricultural Engineering, College of Agricultural and Marine Sciences, Sultan Qaboos University, Al-Khoud 123, Oman
4   Department of Agronomy, Faculty of Agriculture and Environment, The Islamia University of Bahawalpur, Bahawalpur 63100, Pakistan
*   Correspondence: abdurehmanuaf@gmail.com or a.rehman@iub.edu.pk

**Abstract:** Rapidly increasing salinization of arable land is a major threat to crop production globally, and the soil of regions with arid environments, such as Oman, are more prone to this menace. In this work, two complementary studies were carried out to evaluate the effect of soil amendments on soil physicochemical properties and growth of cucumber seedlings. In the first study, high- and low-saline soils were used with or without perlite. The amendments tested included mango wood biochar, silica, and biochar + silica, while no amendment was taken as the control. The second study included two cucumber cultivars and irrigation water with two salinity treatments, along with the same four soil amendments. The results showed that soil amendment with biochar alone or with silica enhanced the soil organic matter and $NO_3$, P, and K concentration, while silica amendment substantially enhanced the soil Si level in both studies. Saline soil and irrigation water inhibited seedling emergence and plant growth in both experiments. However, the addition of biochar and silica alone or in combination increased the cucumber seedling dry weight from 39.5 to 77.3% under salt stress compared to the control. Likewise, silica and biochar + silica reduced the sap Na accumulation by 29–31.1% under high salinity. Application of biochar under high salinity resulted in 87.2% increase in sap K. Soil amendments with biochar and silica or their combination have the potential to reduce the adverse effect of salt stress on cucumber.

**Keywords:** salinization; biomass production; nutrient; irrigation water salinity; mango wood biochar



## 1. Introduction

Soil salinity is one of the leading environmental constraints limiting agricultural productivity in many regions of the world. Globally, about 0.8 billion ha of land is considered as saline [1]. Secondary salinization, which is human-induced salinization, affects more than 75 million ha [2]. Every single year, up to 1.5–2 million ha of land around the world is lost due to salinity [2], and the monetary loss in the agriculture sector due to soil salinity is more than USD 27.3 billion [3]. The majority of irrigated land worldwide is saline compared to nonirrigated land [2,4,5]. The accumulation of salt in the soil produces a high soil solution osmotic pressure, which reduces water availability to plants, leading to wilting and poor growth. Saline-stress-induced deleterious effects on plant growth include photosynthesis reduction due to limited chlorophyll biosynthesis [6], impaired photosynthesis machinery [7], reduced osmotic potential, nutrient imbalance, specific ion toxicity, or a combination of all these factors [8–10]. Salt stress also enhances the accumulation of reactive oxygen species, leading to loss of membrane integrity and electrolyte leakage [11].

Biochar (BC) is a carbon-rich material produced by pyrolysis of biomass under low oxygen supply. Its application can improve plant growth under salt stress conditions as it reduces the soil bulk density, electrical conductivity, and exchangeable $Na^+$ and $Cl^-$ ions in saline soil [12]. For instance, application of maple (*Acer pseudoplatanus* L.) residue BC (50 and 100 g $kg^{-1}$ soil) in saline soils was found to reduce $Na^+$ uptake and reactive oxygen species generation in root and leaves and increase the cation exchange capacity of soil, chlorophyll content index, leaf area, and nutrient uptake in mung bean (*Vigna radiata* (L.) R. Wilczek) [13]. Although silicon (Si) is a nonessential plant nutrient, it still plays a positive role in the growth and development of plants. Increased Si uptake can improve the ability of plants to grow under suboptimal growth environments [14]. Adequate Si supply can help ameliorate the adverse effects of NaCl on plants through activation of signaling molecules and regulation of phytohormones under salt stress conditions. For instance, jasmonic acid signaling was found to upregulate the genes involved in Si uptake under salt stress conditions, which accelerated the antioxidant defense system and osmolyte production under salt stress conditions [15]. In addition to the role of Si in the activation of signaling molecules and regulation of phytohormones, it has been shown to effectively reduce degradation of photosynthetic pigments, improve gas exchange traits, inhibit lipid peroxidation and leaf electrolyte leakage, and enhance osmotic adjustment of plants. The application of Si has been found to increase $K^+$ concentration and reduce $Na^+$ absorption, transport, and accumulation in plants [16].

The Sultanate of Oman is located in an arid region where the annual rainfall is less than 100 mm, resulting in the majority of land being unsuitable for agriculture without irrigation [17]. The main factor contributing to the issue of salinity is the high-salt content of the groundwater in Oman, especially in the Al-Batinah coastal region, where seawater intrusion into the aquifers has increased salinity levels [18]. This has resulted in serious consequences for agriculture, with limited water resources exacerbating the problem [19].

There are several reports regarding the role of BC and Si in abiotic stress tolerance in plants. However, to the best of our knowledge, very little information is available regarding the individual and interactive effects of BC and Si application in cucumber. Cucumber is a widely grown commercial greenhouse crop in Oman. The present study was carried out to evaluate the individual and interactive effects of soil BC and Si amendments on soil physicochemical properties, growth, and nutrient dynamics of cucumber seedlings under conditions of high and low soil electrical conductivity (EC) and high and low salt levels in irrigation water.

## 2. Materials and Methods

### 2.1. Experiment 1

Two types of soils were used. One was from a greenhouse at a local farm with a relatively high EC level due to past fertilization practices (EC ~5.0 dS $m^{-1}$, $Na^+$ ~500 ppm), and the other was from a nonfarmed deposit of soil from Sohar farm with low salt and nutrient content (EC ~1.4 dS $m^{-1}$, $Na^+$ ~160 ppm). The pH, EC, and water-soluble $Na^+$ were measured in a 1:2 (30 g:60 mL) mixture of soil and distilled water according to standard methods [20] using a hand-held calibrated meter (EUTECH, OAKTON, 35425-10). Each of these two soils were used as 100% soil or mixed with commercial-grade perlite in a 1:1 ratio (*v/v*) to improve drainage. Then, in each of the four soil mixtures, four basic treatments were compared—control, BC, Si, and a mixture of BC and Si—that were added at a rate of 10% by volume to each pot and mixed well. All treatments were replicated 3 times for a total of 48 pots. Plastic pots of 8 cm diameter and a volume of 470 mL were used.

In Experiment 1, four cucumber seeds (*Cucumis sativus*) of the variety Jabbar F1 were planted in each pot and watered with tap water (EC = 1.19 dS $m^{-1}$; pH = 7.2) to saturation. Hand watering was performed twice per week as needed, and care was taken to only apply the amount that could be absorbed by the soil to minimize leaching.

Biochar was produced at the AES (Agricultural Experiment Station) at SQU using local dried mango wood. The wood was burned using prototype BC apparatus, and the final

product was crushed and sieved to 2.0 mm as detailed in our previous study [21]. The Si used in this experiment was manufactured by Agripower®, a company based in Australia (https://agripower.com.au/agrisilica-granular-fertiliser/, accessed on 1 March 2022), and was added as off-white granules, which dissolved easily when water was added.

The experiment was conducted in a growth chamber under controlled environmental conditions. After planting, pots were placed on four clear plastic trays in a completely randomized design on four shelves of a growth chamber/incubator with vertical fluorescent lighting on the front, back, and sides. To avoid the effect of distance to light, the pots were frequently rotated within each tray, and the trays were rotated from shelf to shelf. No effect of position within the chamber on plant growth was observed in this or subsequent experiments. The growth chamber (SANYO, MLR-350HT) was set at a continuous temperature of $25/20 \pm 2$ °C (day/night) and 50% humidity. Photosynthetic active photon flux of 380 μmol $s^{-1}$ $m^{-2}$ was measured using a Hydrofarm LGBQM Quantum PAR meter at the surface of the pots, and a light/dark photoperiod (16/8 h) was maintained during the experiment.

### 2.2. Experiment 2

The nonfarmed low-fertility, low EC soil from Experiment 1 was used for all treatments in this experiment, mixed at a ratio of 3:1 with commercial-grade perlite. Two salt treatments were applied to the water in order to simulate a farm with saline irrigation water. Two cucumber genotypes (SV8975CB and Jabbar F1) were compared, and the same soil amendments were compared (no amendment, BC, Si, and BC + Si) at the same rate of 10% by volume in the same sized pots with 3 replications for a total of 48 pots.

Two seeds were planted in each pot, which were thinned to one plant per pot and watered with tap water for a week before exposing them to two salt levels: high-salt water (EC ~3 dS $m^{-1}$) and low-salt water (EC ~0.5 dS $m^{-1}$). Plants were watered with NaCl salt solution every 2 days, and the EC level of 3.0 dS $m^{-1}$ was chosen to stress the cucumber seedlings but not kill them [22]. A balanced soluble nutrient solution was used once a week to improve fertility. After each watering session, the plant trays were rotated on growth chamber shelves to obtain more even light effects. Both experiments were conducted for approximately 5.5 to 6 weeks in the spring of 2018.

### 2.3. Plant Observations

In Experiment 1, the number of emerged seedlings were recorded twice a week, and the emergence percentage was calculated. A vigor rating of 1 (poor growth, weak stem, and few leaves) to 5 (excellent growth, strong stem, and many leaves) was used as visual observation of plant health, and an index was created combining germination × vigor. In Experiment 2, plant height was recorded before harvesting, and the number of leaves was counted on the fifth week after emergence. The chlorophyll density was measured in the fifth week using a SPAD meter (SPAD-502Plus). In both experiments, plants were harvested in week 5.5 or week 6. The root and shoots were separated and weighed immediately for fresh weight using an electric balance. The harvested plant samples were then oven dried at 60 °C for 48 h to dry weight.

After the fresh weight measurements, sap from the shoots was extracted using a garlic press and collected on a small tissue paper to determine the sap nutrient concentration. The concentration of $Na^+$, $K^+$, and $NO_3^-$ in plant sap was analyzed using different meters (HORIBA B-722 for $Na^+$, HORIBA B-731 for $K^+$, and HORIBA B-743 for $NO_3^-$) that were calibrated with nutrient solutions provided by the manufacturer.

### 2.4. Soil Analysis

Phosphorus levels were determined using the "Olsen method" of phosphorus extraction using 5 g of soil and $NaHCO_3$ solution according to [23]. Water-soluble nutrient concentrations were determined in a 1:2 solution of soil and distilled water (*v/v*) and in-

cluded tests for pH, EC, and water soluble $Na^+$, $NO_3$, and $K^+$ using the hand-held meters previously described [24]

The water-holding capacity of each soil was estimated by saturating 30 mL of soil in a filter paper in a funnel. After six hours, the soil was weighed and the saturated soil samples were oven dried for 48 h at 80 °C to a constant value [25]. Then, the water content was calculated as water held in the soil divided by soil dry weight. The organic matter was estimated in soil via loss on ignition (LOI) [26] in a muffle furnace for 2 h as 450 °C.

The concentration of soluble Si in soil was measured according to the methods described in ICARDA [23]. The most abundant form of Si is monomeric silica acid ($H_3SiO_4$), which can be extracted with 0.01 M $CaCl_2$ solution, even in soils with a high level of $CaCO_3$. The absorbance of blank, standards, and samples were recorded on a SPECTRONIC 200E spectrophotometer (Thermo Scientific) at 660 nm. The calibration curve for the standards was prepared, and the absorbance was plotted against the respective Si concentration to convert absorbance to soil concentration.

*2.5. Data Analysis*

The experimental data were analyzed using analysis of variance at $p < 0.05$ using Minitab (Minitab® 17.3.1) and the PROC GLM procedure. Significant treatment means were separated using Tukey's honestly significant difference (HSD) test. Microsoft Excel program was used to develop the figures and calculate standard errors.

**3. Results**

*3.1. Experiment 1*

3.1.1. Soil Physicochemical Properties

The soils used in this experiment (S) significantly ($p < 0.001$) differed for all studied soil physicochemical traits. However, the soil amendment treatments (T) were significantly different for soil organic matter (OM), pH, $NO_3$ ($p < 0.05$), P ($p < 0.001$), $K^+$ ($p < 0.05$), and Si ($p < 0.001$) concentration in soil (Table 1). The interaction S × T was significant ($p < 0.001$) for pH, EC, P, Na, and Si (Table 1). The water-holding capacity (WHC) was higher in the high EC soil obtained from the local greenhouse compared to the never-cropped low EC soil. Likewise, OM was higher in the high-saline soils and also increased by BC application alone or in combination with Si (Table 1). The high EC greenhouse soils also had the highest concentrations of $NO_3$, P, K, Na, and Si compared to the uncropped soil (Table 1).

**Table 1.** Soil nutrient values and significance levels in Experiment 1.

| Treatments<br>Soil Salinity (S) | WHC<br>(%) | OM<br>(%) | pH | EC<br>(dS m$^{-1}$) | NO$_3$<br>(ppm) | P<br>(ppm) | K<br>(ppm) | Na<br>(ppm) | Si<br>(ppm) |
|---|---|---|---|---|---|---|---|---|---|
| High-salt soil | 49.4 B | 12.8 A | 6.9 B | 3.6 A | 253 A | 29 A | 435 A | 415 A | 47 A |
| High-salt soil + perlite | 58.4 A | 11.6 A | 6.8 B | 2.0 B | 167 B | 27 B | 208 B | 230 B | 41 B |
| Low-salt soil | 21.1 D | 3.4 B | 7.5 A | 0.4 C | 92 C | 4 C | 10 C | 59 C | 14 D |
| Low-salt soil + perlite | 29.6 C | 4.0 B | 7.5 A | 0.3 C | 99 C | 4 C | 13 C | 68 C | 17 C |
| $p < 0.05$ | ** | ** | ** | ** | ** | ** | ** | ** | ** |
| Treatments (T) | | | | | | | | | |
| Control | 36.3 | 6.0 B | 7.2 AB | 1.5 | 125 B | 14 C | 158 AB | 179 | 24 B |
| BC | 40.3 | 8.8 A | 7.3 A | 1.6 | 155 AB | 17 B | 193 A | 186 | 25 B |
| Si | 38.9 | 7.2 B | 7.2 AB | 1.5 | 150 AB | 13 C | 133 B | 196 | 36 A |
| BC + Si | 42.8 | 9.7 A | 7.1 B | 1.7 | 181 A | 21 A | 182 AB | 211 | 35 A |
| $p < 0.05$ | ns | ** | * | ns | * | ** | * | ns | ** |
| S × T | ns | Ns | ** | ** | ns | ** | ns | ** | ** |

Means sharing the same letters in the column do not differ significantly at $p < 0.05$. * = $p < 0.05$; ** = $p < 0.01$. WHC = water-holding capacity; OM = organic matter; EC = electrical conductivity; BC = biochar; Si = silica; BC + Si = biochar + silica.

The BC treatments resulted in significantly higher levels of P, and the Si treatments resulted in higher levels of soil Si. Nitrate was highest in the BC plus Si treatment but was only significantly different from the control.

Looking more closely at the interaction effects, the saline soils had lower pH compared to low-saline soils, probably as a result of the fertilization history. In this regard, the lowest soil pH (6.47) was observed for high-saline soil amended with Si + BC, while low-saline soils amended with BC had the highest soil pH (Figure 1a). The soil EC was largely influenced by the soil salinity level, and perlite addition reduced the soil EC. The highest soil EC was recorded for saline soil without perlite (3.75 dS m$^{-1}$), while low-saline soil exhibited the lowest value of EC (0.34 dS m$^{-1}$) irrespective of soil amendment (Figure 1b). Phosphorus level was higher in high-saline soils. BC + Si application in high-saline soil with perlite had the highest soil P concentration (44.1 ppm), while the lowest P concentration (2.66 ppm) was recorded in soils with low salt irrespective of soil amendment (Figure 1c). Soil Si levels were highest overall in the high EC greenhouse soils, probably due to prior application of compost and/or peat moss, which also contains high levels of soluble Si. Application of Si amendment further increased the Si concentration (53.9 ppm) for both Si application alone or in combination with BC. The lowest soil Si concentration was measured for the control (9.8 ppm) and BC amendment (10.2 ppm) in low-saline soils (Figure 1d). The Na concentration was highest in the high EC soils without perlite irrespective of soil amendment and lower when perlite was added (Figure 1e).

### 3.1.2. Germination, Growth, and Sap Nutrient Concentrations

The analysis of variance revealed that soil salinity significantly ($p < 0.001$) affected all the studied traits of cucumber in Experiment 1 (Table 2). However, the four treatments were not significant, except for sap K concentration. The soil salinity by treatment (S × T) interaction was only significant for the sap NO$_3$ concentration (Table 2). All soil types exhibited similar germination rates except the high-saline soil without perlite. Higher vigor ratings and index were noted for all soils except for the high-saline soil without perlite (Table 2). The plants grown in high-saline soils could not survive after germination, and the BC and Si treatments did not reduce the effect of salt enough to increase germination or survival. The highest root and top fresh weight were noted for low saline + perlite and high saline + perlite soils, respectively. Again, there was no significant effect of BC or Si treatments on root or shoot fresh weight (FW) (Table 2). The sap Na and K concentration was highest in plants grown on high saline + perlite soil, while low salinity reduced sap Na and K accumulation. Application of BC alone and BC + Si enhanced sap K ac-cumulation in cucumber (Table 2). In the case of sap NO$_3$, application of Si substantially enhanced NO$_3$ accumulation (690 ppm) in the high salt + perlite soil, while none of the soil amendments enhanced NO$_3$ uptake in the low-salt stress condition (Figure 1f).

**Table 2.** Germination, plant vigor, FW, sap nutrient concentration, and significance levels in Experiment 1.

| Treatments<br>Soil Salinity (S) | Final Germination Count | Average Vigor (0–5 Rating) | Index | Root FW (g Plant$^{-1}$) | Tops FW (g Plant$^{-1}$) | Na (ppm) | NO$_3$ (ppm) | K (ppm) |
|---|---|---|---|---|---|---|---|---|
| High-salt soil | 0.5 B | 0.125 B | 0.125 B | 0 | 0 | 0 | 0 | 0 |
| High-salt soil + perlite | 3.17 A | 0.770 A | 2.562 A | 0.44 B | 1.85 A | 1609 A | 475 A | 2787 A |
| Low-salt soil | 2.75 A | 0.687 A | 2.354 A | 0.54 B | 0.99 B | 626 B | 192 B | 1829 B |
| Low-salt soil + perlite | 3.416 A | 0.854 A | 2.979 A | 1.15 A | 1.11 B | 905 B | 173 B | 1715 B |
| *p* < 0.05 | *** | *** | *** | *** | *** | *** | *** | *** |
| Treatments (T) | | | | | | | | |
| Control | 2.583 | 0.541 | 1.916 | 0.53 | 1.15 | 951 | 228 | 1446 B |
| BC | 2.583 | 0.625 | 1.958 | 0.84 | 1.33 | 991 | 283 | 2877 A |
| Si | 2.50 | 0.625 | 2.00 | 0.74 | 1.48 | 1176 | 346 | 1517 B |
| BC + Si | 2.166 | 0.645 | 2.145 | 0.73 | 1.31 | 1070 | 261 | 2600 A |
| *p* < 0.05 | Ns | ns | ns | ns | ns | ns | ns | *** |
| S × T | Ns | ns | ns | ns | ns | ns | * | ns |

Means sharing the same letters in the column do not differ significantly at $p < 0.05$. * = $p < 0.05$; *** = $p < 0.001$. FW = fresh weight; index = germination × vigor; BC = biochar; Si = silica; BC + Si = biochar + silica.

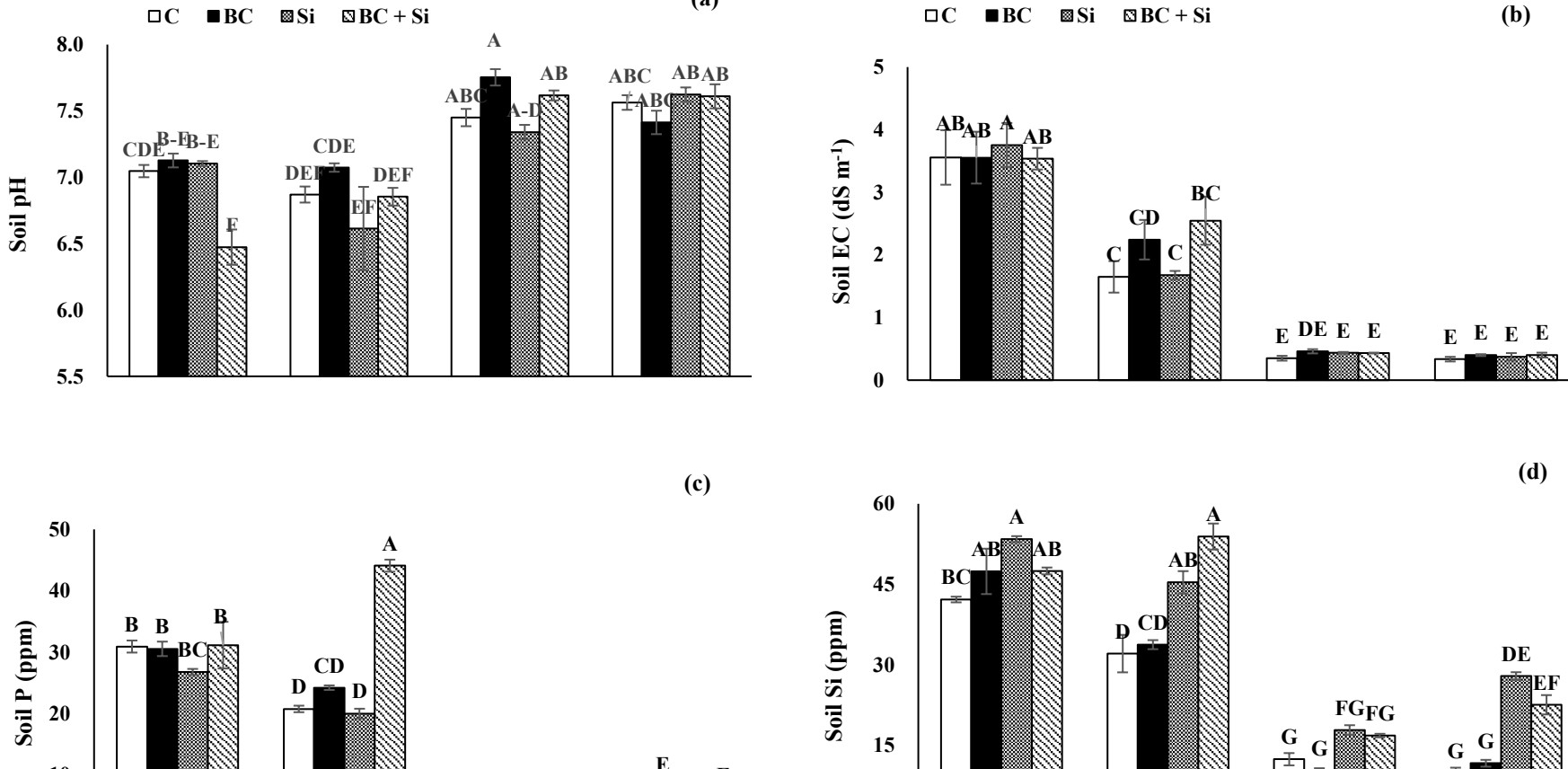

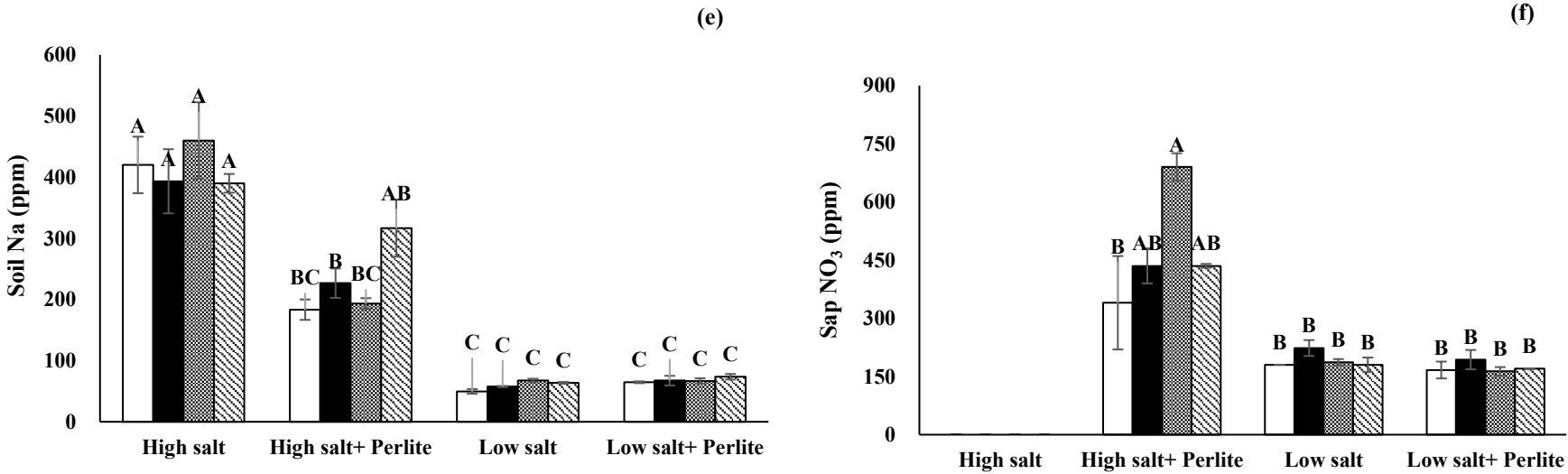

**Figure 1.** Effect of soil type and soil amendment on soil (**a**) pH, (**b**) EC, (**c**) phosphorus, (**d**) Si, (**e**) Na, and (**f**) plant sap $NO_3$ concentrations of cucumber $\pm$ S.E. C = control; BC = biochar; Si = silica; BC + Si = biochar + silica. Means sharing the same letters do not differ significantly at *p* < 0.05.

*3.2. Experiment 2*

3.2.1. Soil Physicochemical Properties

The analysis of variance revealed that irrigation water salinity significantly influenced all soil physicochemical properties except Si concentration (Table 3). Soil amendment treatments also influenced all traits except EC. The interactions S × T were only significant for WHC, pH, and soil K and Na concentrations (Table 3).

**Table 3.** Soil nutrient values and significance levels for Experiment 2.

| Treatments | WHC (%) | OM (%) | pH | EC (dS m$^{-1}$) | Si (ppm) | K (ppm) | Na (ppm) | NO$_3$ (ppm) |
|---|---|---|---|---|---|---|---|---|
| **Irrigation Water Salinity (S)** | | | | | | | | |
| High salt | 38.9 B | 3.18 A | 8.37 B | 2.26 A | 19.1 | 35.3 A | 395 A | 140.7 B |
| Low salt | 36.9 A | 2.91 B | 8.56 A | 0.85 B | 20.4 | 25.7 B | 128 B | 208.8 A |
| $p < 0.05$ | * | * | ** | ** | ns | ** | ** | ** |
| Treatments (T) | | | | | | | | |
| Control | 33.8 C | 2.24 B | 8.30 B | 1.43 | 14.1 B | 12.9 C | 260 AB | 165.9 B |
| BC | 39.3 AB | 3.58 A | 8.64 A | 1.50 | 16.5 B | 58.8 A | 240 B | 191.7 A |
| Si | 37.1 B | 2.54 B | 8.32 B | 1.65 | 23.3 A | 9.2 C | 292 A | 162.4 B |
| BC + Si | 41.5 A | 3.82 A | 8.60 A | 1.65 | 25.0 A | 41.0 B | 253 B | 179.0 AB |
| $p < 0.05$ | ** | ** | ** | ns | ** | ** | * | * |
| S × T | * | ns | * | ns | ns | * | * | ns |

Means sharing the same letters in the column do not differ significantly at $p < 0.05$. * = $p < 0.05$; ** = $p < 0.01$; WHC = water-holding capacity; OM = organic matter; EC = electrical conductivity; BC = biochar; Si = silica; BC + Si = biochar + silica.

Soil amendments improved the WHC, with the highest WHC (43.2%) in the BC + Si treatment, followed by BC alone and then Si alone. The lowest WHC was in the control treatment (33.4%). The higher salinity irrigation treatment seemed to have higher WHC in both BC soil treatments (Figure 2a). Biochar application alone or in combination with Si increased the soil pH (8.70) irrespective of soil salinity level. However, the lowest soil pH was noted for the control (8.26) and Si treatment (8.10) for high-salinity irrigation treatment (Figure 2b). High-salinity irrigation treatment exhibited higher OM accumulation, while BC addition alone or in combination with Si enhanced (3.95%) the soil OM level (Table 3). In the case of soil EC, high-salinity irrigation had the highest EC, while none of the soil amendment treatments significantly influenced the soil EC (Table 3). Application of Si alone or in combination with BC substantially increased (26.4 ppm) the soil Si concentration (Table 3). Low-salinity irrigation resulted in the highest NO$_3$ level. Among soil amendments, BC application augmented the soil NO$_3$ level (226.7 ppm), while Si application and no soil amendment had the lowest NO$_3$ concentration (Table 3). Biochar application substantially augmented the soil K concentration as the highest K level (66.3 ppm) was recorded for high-salinity irrigation treatment receiving BC amendment, while soil receiving Si and no amendment (11.5 ppm) had the lowest K irrespective of irrigation salinity level (Figure 2c). The soil Na level was lowest in the low-salinity irrigation treatment, irrespective of soil amendment. However, in high-salinity irrigation, BC application alone or in combination with Si reduced (122 ppm) the soil Na concentration (Figure 2d).

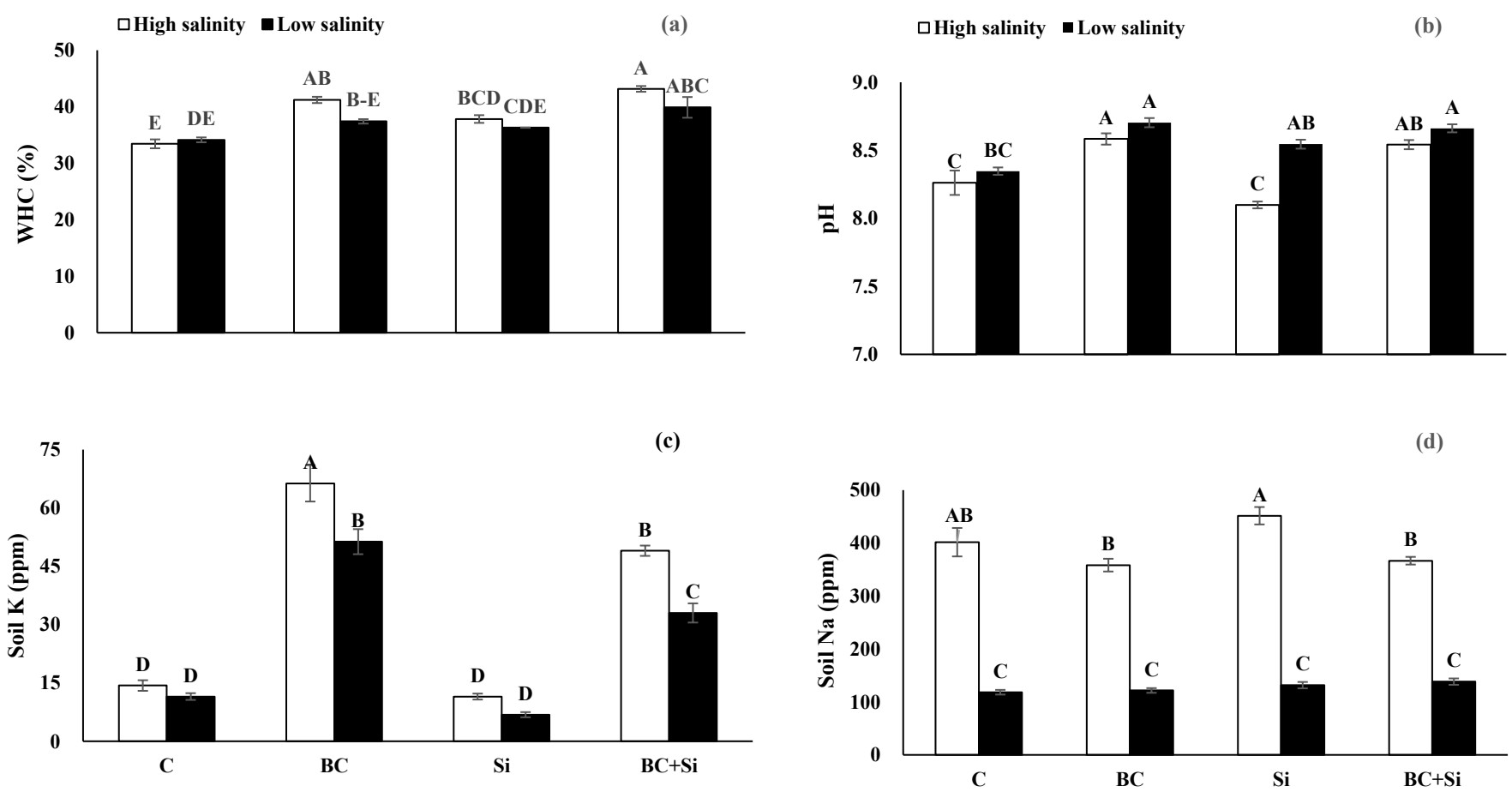

**Figure 2.** Effect of salt level and soil amendments on soil (**a**) water-holding capacity (WHC), (**b**) pH, and (**c**) K and (**d**) Na concentrations ± S.E. C = control; BC = biochar; Si = silica; BC + Si = biochar + silica. Means sharing the same letters do not differ significantly at *p* < 0.05.

### 3.2.2. Biomass, Growth, and Sap Nutrient Concentrations

The analysis of variance revealed that cucumber cultivars (C) significantly differed for leaf count, plant fresh weight, sap $NO_3$, sap Na, and sap K concentration (Table 4). Irrigation salinity level significantly influenced all traits except SPAD value and $NO_3$ concentration in plant tissues. Soil amendments significantly influenced the measured traits. The interaction of C × S was significant only for leaf number and sap K concentration. The C × T interaction was significant for plant height and leaf K concentration. However, the S × T interaction was significant for all traits except plant height and SPAD value. The three-way interaction C × S × T was only significant for leaf count and sap K concentration (Table 4).

**Table 4.** Plant height, leaf count, SPAD, biomass, and sap nutrient content and significance levels for Experiment 2.

| Treatments | Plant Height | Leaf Count | SPAD | Plant fresh Weight | Plant Dry Weight | Leaf NO$_3$ | Leaf Na | Leaf K |
|---|---|---|---|---|---|---|---|---|
| **Cucumber Cultivars (C)** | **(cm)** | | | **(g Plant$^{-1}$)** | **(g Plant$^{-1}$)** | **(ppm)** | **(ppm)** | **(ppm)** |
| SV8975 CB | 6.87 | 3.21 B | 34.3 | 3.11 B | 0.349 | 362 B | 3063 A | 3208 A |
| Jabbar, F1 | 6.48 | 4.05 A | 35.5 | 3.84 A | 0.357 | 511 A | 2734 B | 2447 B |
| *p* < 0.05 | ns | *** | ns | *** | ns | ** | * | *** |
| **Salinity level (S)** | | | | | | | | |
| High salt | 6.01 B | 3.11 B | 34.7 | 2.38 B | 0.248 B | 406 A | 4808 A | 2697 B |
| Low salt | 7.33 A | 4.16 A | 35.1 | 4.56 A | 0.457 A | 467 A | 988 B | 2958 A |
| *p* < 0.05 | *** | *** | ns | *** | *** | ns | *** | * |
| **Treatments (T)** | | | | | | | | |
| Control | 7.03 AB | 3.15 C | 36.5 AB | 2.41 C | 0.262 B | 670 A | 3430 A | 1954 C |
| BC | 7.33 A | 3.88 A | 30.1 C | 4.24 A | 0.410 A | 193 B | 3235 A | 4016 A |
| Si | 6.40 BC | 3.56 B | 39.9 A | 3.24 B | 0.334 AB | 648 A | 2599 B | 1971 C |
| BC + Si | 5.96 C | 3.95 A | 33.1 BC | 3.99 A | 0.404 A | 235 B | 2329 B | 3550 B |
| *p* < 0.05 | *** | *** | *** | *** | *** | *** | *** | *** |
| **Interactions (*p* < 0.05)** | | | | | | | | |
| C × S | ns | * | ns | ns | ns | ns | ns | *** |
| C × T | * | ns | ns | ns | ns | ns | ns | *** |
| S × T | ns | ** | ns | *** | ** | ** | *** | *** |
| C × S × T | ns | * | ns | ns | ns | ns | ns | *** |

Means sharing the same letters in the column do not differ significantly at $p < 0.05$. * = $p< 0.05$; ** = $p < 0.01$; *** = $p < 0.001$. SPAD = soil plant analysis development; BC = biochar; Si = silica; BC + Si = biochar + silica.

The cultivars and irrigation treatment did not influence the SPAD values. However, among the soil amendments, the highest SPAD value was noted with Si application, while the lowest SPAD value was observed with BC application (Table 4). The interaction C × T showed that the tallest cucumber plants (7.63 cm) were noted for cultivar SV8975CB without any soil amendment, while BC + Si combined application reduced (5.87 cm) the plant height in both cultivars (Figure 3a). The highest (five) number of leaves per plant were noted for cultivar Jabbar grown with BC and BC + Si amendment in low-salinity irrigation treatment (Figure 3b). The interactive effect of S × T showed that the highest plant fresh (6.06 g plant$^{-1}$) and dry (0.58 g plant$^{-1}$) weights were produced with application of BC alone or in combination with Si in low-salinity irrigation. The lowest values of fresh (1.74 g plant$^{-1}$) and dry weight (0.17 g plant$^{-1}$) were noted for control plants grown in high-salinity irrigation treatment (Figure 3c,d). The interaction of S × T revealed that highest leaf $NO_3$ was noted with Si application with low-salinity irrigation (810 ppm), which was similar to the control in high-salinity, while the lowest $NO_3$ concentrations was noted for BC and BC + Si in both low (315 ppm) and high (157 ppm) salinity treatments (Figure 3e). High-salinity irrigation substantially increased the Na concentration in plant sap (Table 4). Nevertheless, soil amendments reduced the Na uptake, with Si alone or in combination with BC reducing (742 ppm) the Na accumulation in plant sap (Figure 3f). The interaction C × S × T showed that cultivar SV8975CB had the highest sap K concentration (5367 ppm) under high salinity with BC amendment, while cultivar Jabbar showed the lowest sap K (1100 ppm) under high-saline treatment without BC amendment (Figure 3g).

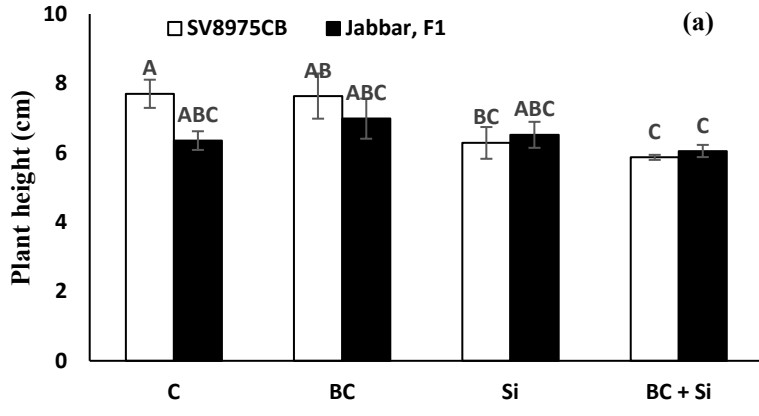

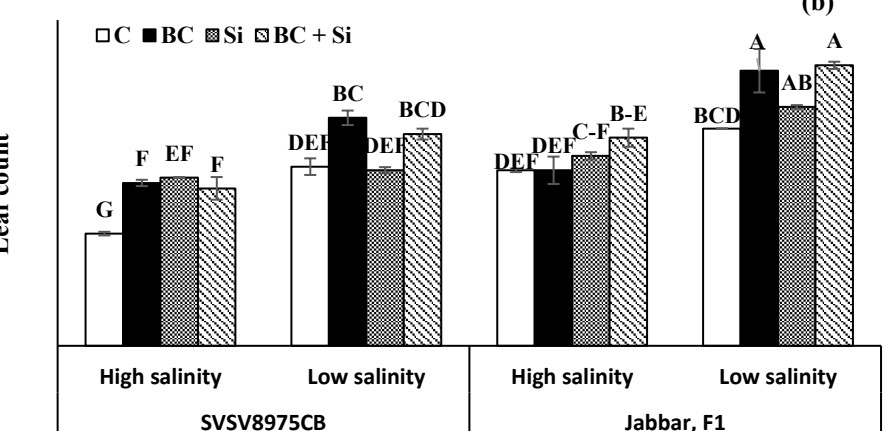

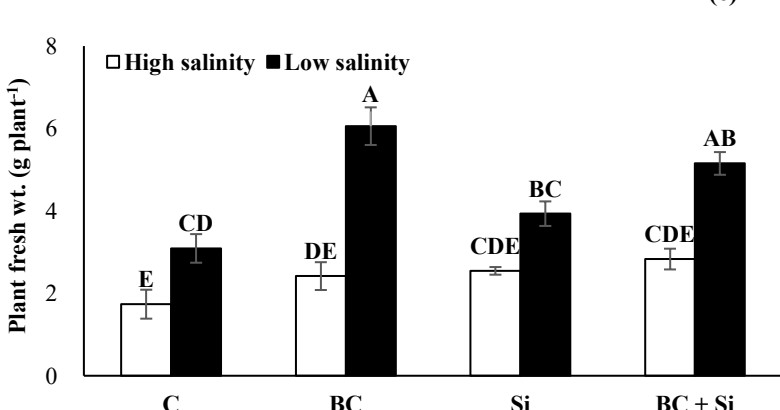

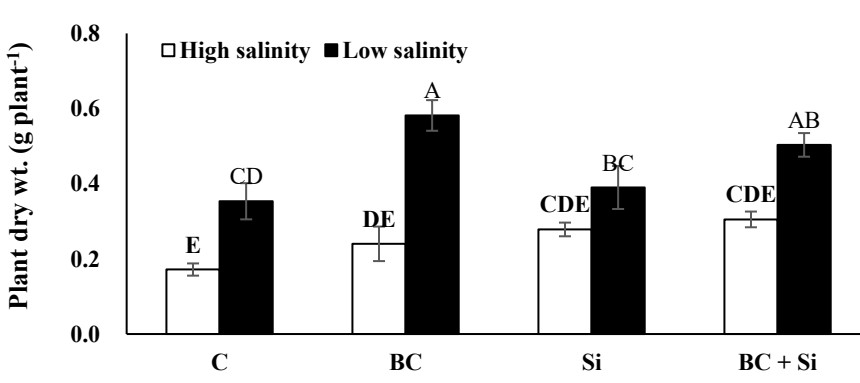

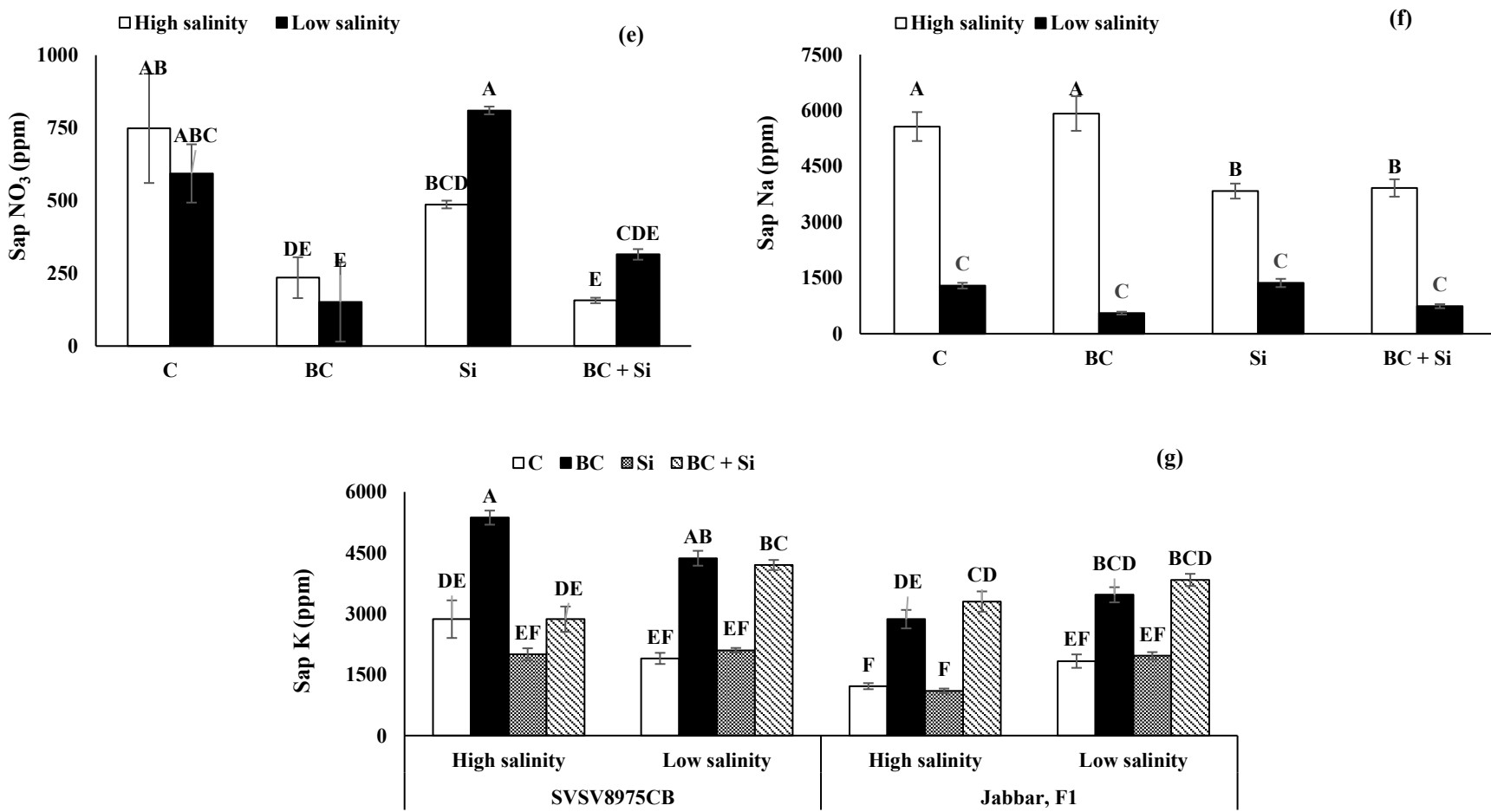

**Figure 3.** Influence of cultivar (C), soil type (S), and soil amendment (SA) interactions on morphology and sap nutrient concentration of cucumber. Comparison of C × SA interaction for (**a**) plant height; C × S × SA interaction for (**b**) leaf count; S × SA interaction for (**c**) plant fresh weight, (**d**) plant dry weight, (**e**) sap $NO_3$ concentration, and (**f**) sap Na concentration; C × S × SA interaction for (**g**) sap K concentration ± S.E. C = control; BC = biochar; Si = silica; BC + Si = biochar + silica. Means sharing the same letters do not differ significantly at $p < 0.05$.

## 4. Discussion

Soil salinization is one of the major threats to crop production globally as it adversely affects soil physicochemical properties and nutrient availability. This study used cucumber as an indicator plant and demonstrated that soil salinity negatively affected the soil properties and plant growth, leading to decreased biomass production as plant growth and yield are greatly contingent on the soil physical properties and nutritional status. However, the addition of soil amendments, such as BC, showed potential in improving plant growth in salt-affected soil.

Soil salinity adversely affected the physicochemical properties in both experiments. In Experiment 1, even though the high EC soil had higher WHC, nutrient levels, and OM due to past fertilization practices, it also had higher EC and soil sodium levels, which negatively affected plant growth. In the first study, the soil OM was enhanced by BC addition, but BC and Si both failed to lower the soluble Na levels of the soil or reduce the uptake of Na measured in the plant sap. Plant growth (germination, vigor, root, or top fresh weight) was not affected by any of the amendments (BC or Si). The OM content was higher in the high-salinity irrigation treatment because it is influenced by two opposing factors, i.e., reduced plant inputs and slower decomposition rate, which could increase soil organic carbon content [27]. However, the addition of BC further enhanced the accumulation of OM in the soil, which can be accredited to the chemical properties of BC, such as the carbon content, as the BC used in this study had higher organic content [21]. Biochar has a porous structure that is more resistant to degradation compared to the original feedstock material [28]. This property allows biochar to significantly improve soil physicochemical and biological properties, which can promote better crop production [29].

In this study, the addition of BC appeared to slightly increase the soluble $NO_3^-$ content of the soil. This is consistent with previous studies that have shown biochar's ability to retain $NO_3^-$ within its pores [30]. However, it should be noted that applying BC to soils may affect the conditions that control nitrification, denitrification [31], and other nitrogen transformation and loss pathways. However, the plant sap levels of $NO_3$ were lower with BC treatment, indicating reduced uptake by plants. This could be due to a complex interaction between the added biochar and other soil components that influence nutrient availability, uptake, and transport in plants. The lower values of soil EC, $NO_3$, P, K, Na, and Si in the less saline soil in Experiment 1 compared to high-saline soil were primarily due to the dilution effect and better fertility of high-saline soil. However, the application of BC in high and low-saline soil increased the soil $NO_3$, P, and K concentrations. This increase in soil nutrient concentration can be attributed to the adsorption of salts and replacement of $Na^+$ from the exchangeable sites in the soil with other cations, such as the $Ca^{2+}$ and $Mg^{2+}$ [32,33]. The biochar's ability to exchange $Na^+$ with these other cations can help to reduce the amount of sodium ions in the soil, which can lead to enhanced leaching of sodium ions from salt-affected soils [32]. Furthermore, biochar can also improve soil fertility by increasing the availability of nutrients to plants through its impact on soil pH. Biochar has been shown to have a liming effect, which can increase soil pH, making nutrients such as P and K more available to plants.

Cucumber is highly sensitive to salt stress, and cucumber seeds sown in high-saline soils showed inhibited germination and reduced vigor. The salinity-induced poor emergence and vigor of cucumber seedling was possibly due to lower activities of germination-related enzymes (e.g., as α-amylase) [21] as lower activities of α-amylase is associated with poor seedling emergence [10,21]. In the present study, root fresh weight was found to be lower with salt treatment and highest in the low-salt soil with perlite. The lower root fresh weight in high-saline soil or control treatment was due to root growth inhibition owing to excessive Na intake [34]. These findings indicate that salt stress disturbs the nutrient homeostasis in plants through excessive accumulation of Na in different parts of plants.

High-salinity irrigation in Experiment 2 decreased plant height, leaf count, plant fresh and dry weight, and sap K, while SPAD and sap $NO_3$ were unaffected by salt level. Sap levels of Na were higher with high-salt treatment but were reduced with

the Si soil amendment treatments. The reduction in Na uptake observed in the Si soil amendment treatments is consistent with the well-established role of Si in enhancing salt stress tolerance in plants as Si application has been shown to improve plant growth and alleviate the negative effects of salt stress by reducing $Na^+$ absorption, improving ion balance, and enhancing osmotic adjustment of the plants [16]. Salt stress disturbs the nutrient homeostasis in plants through excessive accumulation of Na in different parts of plants [1]. However, application of BC enhanced the sap K in both experiments, while Si application substantially reduced the sap Na levels in Experiment 2. Moreover, improvement in plant growth and biomass production with BC amendment can be ascribed to increased availability of nutrients such as N, P, and K from BC itself, which leads to higher levels of K but not $NO_3$ in sap, or modification in nutrient cycling and retention, which improves plant growth [21,35].

Cucumber cultivars also varied in leaf count, plant fresh weight, and tissue nutrient level. The cultivar Jabbar F1 exhibited higher leaf count, fresh weight, and sap $NO_3$, along with lower Na and low K. The variation in these traits may be due to genetic differences and different responses to salt stress as Jabbar restricted Na accumulation, while SV8975CB accumulated more Na in plant tissues. In the present study, the higher leaf count, fresh weight, and sap $NO_3$ levels in Jabbar F1 may be attributed to its ability to maintain better nutrient uptake and assimilation under salt stress conditions. On the other hand, SV8975CB may have accumulated more Na in plant tissues, which may have contributed to its lower growth and nutrient accumulation.

## 5. Conclusions

Soil salinity inhibited the emergence and growth of cucumber. Soil amendments, particularly biochar, improved soil fertility and physical properties such as soil K, P, organic matter content and, in some cases, soil levels of $NO_3$ and water-holding capacity but did not improve plant growth in Experiment 1. When salt stress was imposed as irrigation water, the effect of biochar on plant growth resulted in higher levels of sap K but lower levels of sap $NO_3$ and better plant growth. Silica, but not biochar, reduced plant uptake of Na based on sap Na measurements. The biochar amendment improved plant growth and sap K level under salt stress conditions. Application of biochar and Si can help improve cucumber production on salt-affected soils or under saline irrigation conditions, but there does not appear to be any synergistic effect from applying both at the same time.

**Author Contributions:** Conceptualization, R.R.J., M.M.K. and M.A.; methodology, M.A.-T., R.R.J. and A.R.; software, R.R.J., A.R. and W.M.A.-B.; formal analysis, M.A.-T. and W.M.A.-B.; investigation, M.A.-T.; resources, R.R.J., M.M.K. and M.A.; data curation, R.R.J. and A.R.; writing—original draft preparation, M.A.-T. and A.R.; writing—review and editing, R.R.J. and A.R.; visualization, M.A.-T. and A.R.; supervision, R.R.J.; project administration, W.M.A.-B.; funding acquisition, R.R.J. All authors have read and agreed to the published version of the manuscript.

**Funding:** This research received no external funding.

**Institutional Review Board Statement:** Ethical review and approval were waived for this study because the study did not involve humans or animals.

**Data Availability Statement:** Data from this study will be shared upon reasonable request to the corresponding author.

**Acknowledgments:** The authors would like to thank Sultan Qaboos University for in-kind support for use of the laboratory, growth chamber, and other facilities, which made this work possible.

**Conflicts of Interest:** The authors declare no conflict of interest.

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
