# Peer review of "Silica and Biochar Amendments Improve Cucumber Growth under Saline Conditions"

_soilsystems, doi:10.3390/soilsystems7010026_

Round 1

Reviewer 1 Report

Soil salinity is one of the leading environmental constraints which is limiting agricultural productivity in many regions of the world. The studies were carried out to evaluate the individual and interactive effect of soil biochar and silica amendments on soil physicochemical properties, growth and nutrient dynamics of cumber seedlings under conditions of high and low soil EC levels, and high and low salt levels in irrigation water. The results showed that biochar could increase the content of potassium in cucumber, and silicon and biochar mixed with silicon could reduce the content of sodium in cucumber. Therefore, Soil amendments with biochar and silicon or their combination have the potential to reduce the adverse effect of salt stress on cucumber。The author 's thinking is clear and the monitoring indicators are comprehensive. In this paper, different saline-alkali land, cucumber varieties and soil additives were used as experimental factors to evaluate the effects of soil amendments on soil physical and chemical properties and cucumber seedling growth. But this article still has the following shortcomings. 

1. The purpose and results of the abstract continue to polish to reflect the value of this article.

2. The continuity between the paragraphs of the introduction is too poor, please modify.

3.L82:The unit of EC is missing.

4.L90:Can 4 seeds reflect the germination rate of seeds? Please explain.

5.L205~212:The abbreviation ' FW ' of ' fresh weight ' in the article is not mentioned, and the abbreviation is expressed as fw or Fw in Table 2, please unify, and check the full text of the relevant format issues.

6. The results of the article need to add data description.

7. Please remove the redundant part of the error bar in the article diagram.

7. The analysis of Si in the discussion is missing, please add.

8. The author needs to unify the reference format according to the requirements of the journal, such as font thickening, font, information sorting, etc.

Author Response

Respected reviewer,

We thank for your very useful comments and suggestions, which have significantly improved the above manuscript. We have revised the manuscript and incorporated all the suggestions. The changes have been made in the revised article using track changes. Moreover, our point-by-point response is given below as marked in blue.

Reviewer 2 Report

The authors describe their research about the individual and interactive effect of soil biochar and silica amendments on soil physicochemical properties, growth and nutrient dynamics of cucumber seedlings, which aims at reducing the adverse effect of salt stress on cucumber production. In fact, there have been many related studies on the interactions of biochar and some trace elements in soils, including silica. Meanwhile, the authors’ research methods are very traditional without much novelty. Therefore, the authors have made a comprehensive study with practical significance, but there are many places that the authors could improve upon, especially in the section of Introduction and Results. My detailed comments are as follows

(1) The introduction had six paragraphs but each paragraph was too short. I suggest that the author reduce the number of paragraphs and expand the content of each paragraph.

(2) Line 138:I doubt the accuracy of using different meters to measure the concentration of Na+, K+ and NO3- in plant sap, instead of chemical analysis.

(3) I doubt whether the results of HSD test in this manuscript were accurate. I can’t find a single letter of B on top of the columns in many sub-figures, such as Fig.1 a,b,d, Fig.2 a,b and Fig.3 a-e, which was not statistically right.

(4) I can find many different forms of the same expression in this manuscript. For example, in the tables and figure captions, you can see “p- value”, “P < value”, “p-value ≤ 0.05”, “p < 0.05”, “p ≤ 0.05”, etc.

(5) The format of this manuscript is kind of messy and needs to be improved in typesetting. For example, the space between some character and symbols are not comfortable: “C x S x T” in Line 254, “ofS x T” in Line 263, and the page is mostly empty after Line 275.  

(6) Fig 1, 2 and 3:The figures cannot be spread across pages. I can see both “Figure 1” and “Fig. 1”.

(7) Line 295:The authors said “This study using cucumber as model plant”. Actually, in this study, you only used cucumber as an indicator plant, just like you said in Line 299. As far as I know, arabidopsis is the most commonly used model plant in botany research, and I haven’t known any study that used cucumber as model plant.

Author Response

Respected Reviewer,

We thank reviewers for your very useful comments and suggestions, which have significantly improved the above manuscript. We have revised the manuscript and incorporated all the suggestions. The changes have been made in the revised article using track changes. Moreover, our point-by-point response is given below as marked in blue.

Reviewer 3 Report

The introduction does not provide enough essential information for the purpose of this study. The logic and objectives need to be rearranged. English also needs to be checked.  

1.      Line 37: m you mean million? If so, use M rather than m.

2.      Line 41-42, 44: check the sentence.

3.      Line 50: it you mean biochar? I would rearrange the two sentences and introduce biochar first and then mention its usage in agriculture.

4.      Line 55: Silicon (Si) and then use the acronym afterward except the first ward in a sentence. Make changes accordingly throughput the paper.

5.      Line 62-66: It does not make sense to me that the issue of salinity were coming from hot weather and water resources. Hot weather may contribute a bit, but I would think the ocean salty water would be the major reason. Please make change here and maybe rewrite those four sentences to explain why your study is necessary.

6.      Line 71-72: How is cucumber been commercially produced there? Greenhouse with field soil?  

7.      Line 91: Do you have information about the tap water? pH, EC?

Results:

1.      Table 1: Why in high salt soils, adding perlite decreased NO3, K, Na and Si while in low salt soils, perlite addition ended up increasing Si?

2.      Figures: Many of the data from figures are not significant from each other and I’m not sure why need to include them since you have the table, which included all data, I suppose?

Discussion:

1.      Your discussion is way too short, not enough to explain your findings. You need to extend your discussion section and discuss everything from your results in detail.

 Please see the attached file for more detail.

Author Response

Respected reviewer,

We thank reviewers for your very useful comments and suggestions, which have significantly improved the above manuscript. We have revised the manuscript and incorporated all the suggestions. The changes have been made in the revised article using track changes. Moreover, our point-by-point response is given below as marked in blue.

Round 2

Reviewer 2 Report

It can be accepted in the current version.

Author Response

Reviewer # 2

It can be accepted in the current version

Response: Thanks, for your valuable comments which really improved the quality of our manuscript

Reviewer 3 Report

some of the questions are not answered.

Your discussion is way too short, not enough to explain your findings. You need to extend your discussion section and discuss everything from your results in detail.

Author Response

Reviewer #3

Your discussion is way too short, not enough to explain your findings. You need to extend your discussion section and discuss everything from your results in detail.

Response: Thanks, the discussion  section has been expanded as suggested 
